# Does Vaginal Cuff Creation and Avoidance of a Uterine Manipulator Improve the Prognosis of Total Laparoscopic Radical Hysterectomy for Early Cervical Cancer? A Retrospective Multicenter Study

**DOI:** 10.3390/cancers14184389

**Published:** 2022-09-09

**Authors:** Eiji Kondo, Kenta Yoshida, Michiko Kubo-Kaneda, Masafumi Nii, Kota Okamoto, Shoichi Magawa, Ryo Nimua, Asumi Okumura, Toshiharu Okugawa, Takaharu Yamawaki, Kenji Nagao, Kouichi Yoshimura, Naoki Watashige, Kenji Yanoh, Tomoaki Ikeda

**Affiliations:** 1Department of Obstetrics and Gynecology, Mie University School of Medicine, Tsu 514-8507, Mie, Japan; 2Department of Obstetrics and Gynecology, Japanese Red Cross Ise Hospital, Ise 516-8512, Mie, Japan; 3Department of Obstetrics and Gynecology, Yokkaichi Municipal Hospotal, Yokkaichi 510-8567, Mie, Japan; 4Department of Obstetrics and Gynecology, Mie Central Medical Center, Tsu 514-1101, Mie, Japan; 5Department of Obstetrics and Gynecology, Saiseikai Matsusaka General Hospital, Matsusaka 515-8557, Mie, Japan; 6Department of Obstetrics and Gynecology, Suzuka General Hospital, Suzuka 513-8630, Mie, Japan

**Keywords:** laparoscopic surgery, radical hysterectomy, cervical cancer, vaginal cuff

## Abstract

**Simple Summary:**

We aimed to investigate the relationship between the collapse of the vaginal cuff and the prognosis of TLRH. In this retrospective multicenter analysis, 94 early cervical cancer patients who underwent O-RH or TLRH in six hospitals in Japan between September 2016 and July 2020 were included; 36 patients underwent TLRH. PFS and OS were not significantly different between O-RH and TLRH groups. Tumor spillage was prevented by creating a vaginal cuff to avoid using a uterine manipulator. Therefore, TLRH might be considered efficient.

**Abstract:**

Our goal was to compare the treatment outcomes of open-abdominal radical hysterectomy (O-RH) and total laparoscopic hysterectomy (TLRH) with vaginal cuff creation and without using a uterine manipulator in stage IB1-B2 (tumor size < 4 cm) cervical cancer cases. In this retrospective multicenter analysis, 94 cervical cancer stage IB1-B2 patients who underwent O-RH or TLRH in six hospitals in Japan between September 2016 and July 2020 were included; 36 patients underwent TLRH. Propensity score matching was performed because the tumor diameter was large, and positive cases of lymph node metastases were included in the O-RH group due to selection bias. The primary endpoint was progression-free survival (PFS) and recurrence sites of TLRH and O-RH. PFS and OS (overall survival) were not significant in both the TLRH (*n* = 27) and O-RH (*n* = 27) groups; none required conversion to laparotomy. The maximum tumor size was <2 and ≥2 cm in 12 (44.4%) and 15 (55.6%) patients, respectively, in both groups. Reportedly, the TLRH group had lesser bleeding than the O-RH group (*p* < 0.001). Median follow-up was 33.5 (2–65) and 41.5 (6–75) months in the TLRH and O-RH groups, respectively. PFS and OS were not significantly different between the two groups (TLRH: 92.6%, O-RH: 92.6%; log-rank *p* = 0.985 and 97.2%, 100%; *p* = 0.317, respectively). The prognosis of early cervical cancer was not significantly different between TLRH and O-RH. Tumor spillage was prevented by creating a vaginal cuff and avoiding the use of a uterine manipulator. Therefore, TLRH might be considered efficient.

## 1. Introduction

Cervical cancer is the fourth leading cause of cancer-related deaths in women. In 2020, 604,127 new cases were detected, and 341,831 deaths were reported worldwide [1]. Meanwhile, in Japan, 11,012 new cases of cervical cancer and 2795 deaths were reported in 2017. In 1992, Nezhat et al. reported the first total laparoscopic radical hysterectomy (TLRH) [2] and, in 2010, Wright et al. [3] described the increasing use of TLRH in the United States, accounting for 1.8% of hysterectomy procedures in 2006 to 31% in 2010.

Previous studies have reported a similar prognosis between TLRH and open-abdominal radical hysterectomy (O-RH) [4,5,6]. However, the Laparoscopic Approach to Cervical Cancer (LACC) trial, a randomized phase III trial, was initiated in 2008 by Ramirez et al. [7]. The overall survival (OS) and progression-free survival (PFS) of TLRH were found to be inferior to those of laparotomy, at 93.8% vs. 99% and 86% vs. 96.6%, respectively. The local recurrence rate was significantly higher than that of laparotomy, with a hazard ratio (HR) of 4.26 (95% confidence interval (CI); 1.44–12.6, *p* = 0.009). Melamed et al. [8] reported that minimally invasive surgery (MIS), either laparoscopic or robotic RH, was associated with a significantly poorer outcome than laparotomy; the mortality rate at 4 years was 5.3% vs. 9.1% (HR 1.65 (95% CI; 1.22–2.22, *p* = 0.002)). Therefore, laparotomy is the reference standard for radical hysterectomy for cervical cancer, and MIS is no longer being performed. In this regard, Lewicki et al. [9] reported that before the LACC trial was published, the minimally invasive approach was used in 58.0% of hysterectomies, compared to 42.9% of hysterectomies after its publication (*p* < 0.001).

The poorer prognosis associated with MIS may be explained by the use of uterine manipulators. A uterine manipulator is routinely used for benign gynecological diseases and is widely used for preventing complications and for the associated ease of surgery [10]. However, its use is considered to be a factor in tumor spread and poorer prognosis [9]. Recently, similar reports have emerged for endometrial cancer [11]. After the LACC trial, a systematic review and meta-analysis by Nitecki et al. [12] reported that the pooled hazard of recurrence or death was 71% higher among patients who underwent minimal invasive radical hysterectomy compared with those who underwent open surgery (HR: 1.71; 95% CI, 1.36–2.15; *p* < 0.001), and that the hazard ratio for death was 56% higher (HR: 1.56; 95% CI, 1.16–2.11; *p* = 0.004). In contrast, a meta-analysis by Ronsini et al. [13] reported that laparoscopic-assisted vaginal radical hysterectomy is a safe MIS option and that it does not appear to affect disease-free and OS in early stage cervical cancer patients. The MEMORY study [14] has revealed that minimally invasive radical hysterectomy for cervical cancer did not appear to compromise oncologic outcomes compared to open surgery, achieving similar PFS and OS rates. This may be due to the fact that these MIS procedures were performed by an experienced gynecologic oncologist. Furthermore, Alfonzo et al. [15] reported the safety of MIS for tumors > 2 cm, while Manzour et al. [16] found no differences in patterns of relapse across surgical approaches in patients with stage 1B1 cervical cancer undergoing radical hysterectomy as the primary treatment.

In Japan, TLRH was approved as a highly advanced medical treatment in 2014 and has been covered by health insurance since 2018. We started performing TLRH at our institution in 2016. However, the Japan Society of Obstetrics and Gynecology (JSOG) recommends that tumor spillage be avoided in the abdominal cavity when cutting the vaginal canal and that close attention be paid to the incision method and uterine removal, based on the results of the LACC trial where local recurrence was significantly higher in the MIS group. Several measures, such as vaginal cuff creation [17,18,19] and stapler use [20], have been described to prevent tumor spillage in the abdominal cavity; the cause for which is often unknown. 

To the best of our knowledge, some papers have reported on the efficacy of TLRH for cervical cancer with a tumor size of <2 cm, but no paper has mentioned the efficacy of TLRH for ≥2 cm cervical cancers. In this study, we aimed to investigate the relationship between the collapse of the vaginal cuff and the prognosis of TLRH. Moreover, we sought to identify factors with the greatest impact on surgical treatment of cervical cancer when cuff creation was performed.

## 2. Materials and Methods

The Mie Gynecologic Oncology Survey (MGOS) was approved by the institutional review board (No. H2021-230) and performed according to the ethical standards of the Declaration of Helsinki revised in 2001. This multicenter retrospective study was approved by the Ethics Committee of Mie University Hospital. Informed consent was obtained in the form of opt-out consent on the hospital website.

Data from six hospitals affiliated with Mie University School of Medicine were retrospectively analyzed. The patients diagnosed with cervical cancer stage IB1-2 as per the International Federation of Gynecology and Obstetrics (FIGO 2018) staging between September 2016 and July 2020 and treated surgically were consecutively enrolled. The inclusion criteria were histologically confirmed cases of squamous cell carcinoma, adenocarcinoma, basal cell carcinoma, and adenosquamous carcinoma. The diagnosis was defined based on the magnetic resonance imaging (MRI) and physical examination findings, as per the FIGO 2018 guidelines. There were 195 patients diagnosed with cervical cancer stages IA2, IB1, IB2, IB3, IIA1, IIA2, and IIB who underwent O-RH or TLRH in six hospitals in the study period. We excluded 99 patients who were diagnosed with cervical cancer stages IA2, IB3, IIA1, IIA2, and IIB. The surgeries performed included type III radical hysterectomy and pelvic lymphadenectomy by laparotomy and laparoscopy. Of the 94 patients with stage IB1 and IB2 identified, 36 who underwent TLRH at Mie University Hospital and 58 who underwent O-RH in six hospitals, including Mie University Hospital, were evaluated. Propensity score matching (PSM) was performed because the tumor diameter was large, and many lymph node metastasis-positive cases were included in the O-RH group owing to selection bias. After PSM, data of 27 TLRH and 27 O-RH cases were compared and examined. The primary endpoint was PFS and recurrence rate with TLRH or O-RH. We compared the intra- and post-operative complications and prognostic factors using the following criteria: histology, age, and surgical procedure.

The choice of adjuvant therapy depended on the institutional criteria. Adjuvant external beam radiotherapy was performed at a dose of 50.4 Gy to the entire pelvis. Adjuvant chemotherapy consists of a combined regimen of paclitaxel 175 mg/m^2^ q3w and carboplatin area under the curve (AUC) 5 q3w.

### 2.1. Clinical and Pathologic Data

Clinical and pathologic data were procured from the medical records of patients. The data collected were age, body mass index (BMI), history of nulliparity, tumor diameter by MRI, surgical procedures, intra- and post-operative complications, operative time, amount of bleeding, number of harvested lymph nodes, lymph node status, post-operative therapy, post-operative FIGO stage, and prognostic information. Intraoperative complications included major bowel, bladder, ureteral, vessel, and nerve injuries. Post-operative complications (ileus, lymphangitis, lymphocyst formation, and venous thromboembolic events) within 30 days of surgery were recorded. The operative time of TLRH was defined as the time from cuff creation to skin closure, whereas that of O-RH was from incision to skin closure. Pathologic data included histology subtype, lympho-vascular involvement, tumor diameter by histology, lymph node metastasis, and parametrial invasion. Finally, we also investigated the incidence and site of the collapse of the vaginal cuff.

### 2.2. Treatment Protocol and Response Assessment

TLRH was the preferred method as per the Mie University Hospital institutional policy for a tumor size < 2 cm. Other hospitals performed O-RH for all patients with stage IB1. The critical steps in each operation were performed by the main surgeon. In this study, there were 36 cases of TLRH; the main surgeon had an experience with 15 cases of TLRH at another hospital.

In 7 out of 36 patients in the TLRH group, the vaginal cuff was not created after conization because the results of the LACC trial were not published until then. However, a vaginal cuff was created in spite of conization to prevent the spread of an intraperitoneal tumor, given that residual tumor can remain despite conization.

### 2.3. Variable Definitions

We investigated the PFS, OS, adverse events, and follow-up data obtained from the date of initial surgery to the date of the last hospital visit. Recurrence was determined by computed tomography. Adverse events were evaluated according to the Clavien–Dindo classification [21].

### 2.4. Statistical Analysis

Continuous variables were compared using the Mann–Whitney test, while categorical variables were compared using Fisher’s exact test. The Kaplan–Meier method and the log-rank test were used for the OS and PFS analyses. The Cox-proportional hazards model was used to identify independent predictors of PFS in cervical cancer. All statistical analyses were performed using the Statistical Package for the Social Sciences (IBM SPSS, version 27.0, Armonk, NY, USA). A *p*-value < 0.05 was considered statistically significant.

### 2.5. Procedure

The procedural code for radical hysterectomy included resection of the parametrium or uterosacral ligaments, resection of the upper 2–3 cm of the vagina, and pelvic lymphadenectomy, according to Piver–Rutledge–Smith Classification III.

### 2.6. Port Site Setting

The vaginal cuff creation was as described in a previous report [19] (Figure 1a–c).

The vaginal cuff creation technique is shown in Appendix A. After creating the vaginal cuff, the patient was placed in the lithotomy position with the pelvis elevated by 10°–15°. First, a 12 mm trocar was placed through the supra umbilical port. Then, three 5 mm trocars were placed in the suprapubic and bilateral lower quadrants, and a 12 mm trocar (Number 5) was placed in the right-lateral position for insertion of the organ retractor and manipulated. The primary surgeon was positioned on the left side. Then, laparoscopic surgery was performed with the trocars arranged in a 5-port diamond configuration and on the upper right side of the abdomen. We then proceeded to perform pelvic lymph node dissection of the external iliac, internal iliac, common iliac, cardinal, sacral, inguinal, and obturator lymph nodes. Type III radical hysterectomy was done, with the uterus removed using an EZ Pass^®^ (Hakko Medical Device, Tokyo, Japan). Next, lymph nodes were removed en bloc and collected via the 12 mm port using EZ Pass^®^ (Hakko Medical Device Division) via the umbilical port. The blood vessels around the uterus were dissected after being clipped using the standard method, the various ligaments were dissected, and the Douglas’ pouch and peritoneum were incised to expose the hypogastric nerve. Subsequently, the anterior and posterior layers of the vesicouterine ligament were dissected while preserving the bladder branch of the pelvic plexus. Then, the vaginal canal was incised, and the uterus and bilateral adnexa were placed in the EZ Pass^®^ and removed transvaginally. Finally, the vaginal wall was closed with a single 0-VICRYL^®^ (Ethicon US LLC. 4545 Creek Rd #3. Cincinnati, OH, USA) suture.

## 3. Results

### 3.1. Patient Characteristics

The patient characteristics are shown in Table 1.

Conversion to laparotomy was not necessary for any patient. The TLRH and O-RH groups were not significantly different with respect to age, BMI, and histology subtype (*p* > 0.05). On the contrary, there were significant differences in the post-operative stage, tumor size, lymphovascular stromal invasion (LVSI), and lymph node metastasis (*p* < 0.01) post-operatively. The O-RH group had a poorer prognosis than the TLRH group. Therefore, the PSM of 94 patients’ data resulted in a final cohort of 54 patients (27 per group), after balancing for post-operative FIGO stage, tumor size, LVSI, and lymph node metastasis, as shown in Table 2.

The maximum tumor size was <2 cm in 12 (44.4%) patients and ≥2 cm in 15 (55.6%) patients in both groups. We compared the prognosis of the two groups. As previously reported, the amount of bleeding in the TLRH group was lesser than that in the O-RH group (*p* < 0.001).

### 3.2. Survival Outcomes

The median follow-up was 33.5 (2–65) months in the TLRH group and 41.5 (6–75) months in the O-RH group.

The PFS of TLRH was 92.6%, which was comparable to the prognosis of O-RH (92.6%). The prognosis of our related facilities was also similar and was not significantly different between the two groups (log-rank *p* = 0.985, Cox HR: 1.019; 95% CI: 0.144–7.238; *p* = 0.985), as shown in Figure 2. Furthermore, the OS of TLRH and O-RH was not significantly different (log-rank *p* = 0.317) (Figure 3).

Furthermore, the maximum tumor size of <2 cm was diagnosed in 12 (44.4%) patients and >2 cm was diagnosed in 15 (55.6%) in both patient groups in this study. In stage 1B1 cases, if the tumor size is ≥2 cm, the prognosis of TLRH might be considered to be equivalent to that of O-RH (*p* = 0.317).

In addition, the PFS and OS of TLRH (*n* = 36; all patients) are shown in Figure 4a,b, and the PFS and OS of TLRH (<2 cm vs. ≥2 cm) are shown in Figure 5a,b. The prognosis of our related facilities was also similar and was not significantly different between the two groups (log-rank *p* = 0.714, Cox HR: 1.670; 95% CI: 0.104–26.716; *p* = 0.717). When analyzing the pattern of recurrence in the TLRH group, we found one case (FIGO stage IIIC1p) with ascending metastasis of the para-aortic lymph node. However, in the other case (FIGO2018 stage IB1), the tumor was disseminated to the peritoneum. In contrast, two cases of lung metastasis were observed in the O-RH group (FIGO2018 stage IB1 and IIB). 

In Table 3, we describe the five patients who developed a collapsed vaginal cuff during surgery.

### 3.3. Adjuvant Therapy

Adjuvant therapy was administered to 13 out of 27 patients in the TLRH group, and 12 out of 27 patients in the O-RH group. Of the 17 cases, 9 underwent chemotherapy, and 8 received radiotherapy, including concurrent chemoradiotherapy in the TLRH group. There was no significant difference between the two groups (*p* = 1.00).

### 3.4. Adverse Events

Many previous studies have reported complications such as ileus in the laparotomy group [4,5,6]. However, we found no significant differences between the two groups in the incidence of adverse events (*p* > 0.05).

## 4. Discussion

In this study, there was no significant difference in the prognosis of early cervical cancer <4 cm between the TLRH and O-RH groups. Tumor spillage was prevented by creating a vaginal cuff and avoiding the use of a uterine manipulator. TLRH might be considered to be an effective procedure.

TLRH may be effective for early cervical cancer. The 5-year survival rate was 92.4% for patients with stage IB (FIGO 2008) cervical cancer in the Annual Treatment Report for 2012 by the Committee on Gynecologic Oncology, the Japan Society of Obstetrics and Gynecology. Our study findings were similar [22].

Several retrospective studies have reported that MIS (TLRH or robotic radical hysterectomy) and O-RH have comparable results [4,5,6]. Shah et al. [4] reported that there was no significant difference in the 3-year OS between robotic radical hysterectomy (95%, *n* = 202) and O-RH (97.2%, *n* = 109). Corrado et al. [5] reported the prognosis of O-RH, which are as follows: stage IB1 (2018), 88.7% (*n* = 101) vs. 89.7% (*n* = 152; LH: laparoscopic hysterectomy) and 88.8% (*n* = 88; RH: robotic hysterectomy; Type B RH, <2 cm; Type C RH, >2 cm. However, they did not describe techniques to prevent tumor spillage and avoid the use of a manipulator.

Margul et al. compared the outcome of patients with stage IB1 cervical cancer on the National Cancer Database from 2010 to 2013. Although MIS was associated with cost reduction and reduced incidence of surgical complications, patients who underwent MIS with tumor size ≤2 cm had a decreased 5-year survival rate compared to those who underwent O-RH (81.3% vs. 90.8%; *p* < 0.001) [23]. FIGO has recently classified cervical cancer into stages 1B1 and 1B2 when it is ≤2 and ≥2 cm, respectively.

However, the recurrence rate in the TLRH group with a tumor size ≤ 2 or >2 cm was not significantly different (4.7% (1/21) vs. 6.6% (1/15); *p* = 1.00) in our study. Furthermore, the maximum tumor size of <2 cm was diagnosed in 12 (44.4%) patients and >2 cm was diagnosed in 15 (55.6%) in both patient groups in this study. In stage 1B1 cases, if the tumor size is >2 and <4 cm, the prognosis of TLRH might be considered to be equivalent to that of O-RH (*p* = 0.317).

One patient (FIGO stage IIIC1p) with tumor size < 2 cm who experienced recurrence had ascending metastasis of the para-aortic lymph nodes. We believe that if this case was managed by O-RH, one would expect the same recurrence pattern of lymphatic metastasis. There was no difference in distant metastasis between the two groups.

The other patient with tumor size ≥ 2 cm who experienced recurrence had peritoneal dissemination. Tumor spillage cannot be determined to be a direct prognostic factor; however, this case had several other poor prognostic factors. This case was diagnosed histologically as squamous cell carcinoma grade 3, which was undoubtedly one of the causes of recurrence.

We analyzed the five cases in which a collapse of the vaginal cuff occurred, as depicted in Table 3. Two cases had a tumor size < 2 cm, while three cases had a tumor size ≥ 2 cm. Tumor size does not seem to be a poor prognostic factor for the collapse of the vaginal cuff. The site of the vaginal cuff collapse was the anterior vaginal wall in two cases, the right lateral wall in two cases, and the left lateral wall in one case. Three cases had a very minimal collapse of the vaginal cuff, such that the rupture site was only noticeable after removing the specimens, as we could not find it during operation.

During colpotomy, it was difficult to recognize the bilateral ends of the vaginal cuff due to the thickness of the parametrium; thus, the collapse of the vaginal cuff on the lateral side was higher than that on the anterior and posterior vaginal walls. However, there was no recurrence observed in four out of five cases during the follow-up period.

The most important measure for better surgical outcomes is avoidance of a manipulator and prevention of tumor spillage, although TLRH, when performed without collapse of the vaginal cuff, is a good procedure.

In the future, a technique to prevent the collapse of the vaginal cuff by injection of a fluorescent dye into the tumor and visualization of the tumor by laparoscopy is warranted, as is a device that can enable complete recognition of the lower end of the vaginal cuff.

Another strength of this study was the low performance rate of conization (27.7%, 10/36) before TLRH. Sert et al. recently reported a similar prognosis between TLRH without the use of a uterine manipulator and O-RH. However, they performed conization in a larger number of patients (62.4%, 147/229) in their study [24].

This study had several limitations. First, this was retrospective in nature; therefore, it is prone to selection bias. The study design also limited the collection of all variables, such as details of the patients’ quality of life. Second, the sample size was small; therefore, the power of the study may have been inadequate to detect a statistically significant difference, or the type II error may have been high. Third, the follow-up period was limited; therefore, future studies with a larger sample size are warranted, as the number of included patients was small in the present study. Fourth, even if there were no significant differences after PSM, the patients in the laparotomy group clearly had several poor prognostic factors. Finally, the main procedure of TLRH was performed by only one surgeon.

## 5. Conclusions

Tumor spillage was prevented during surgery by the creation of a vaginal cuff and avoidance of a uterine manipulator. TLRH might be considered an effective procedure.

## Figures and Tables

**Figure 1 cancers-14-04389-f001:**
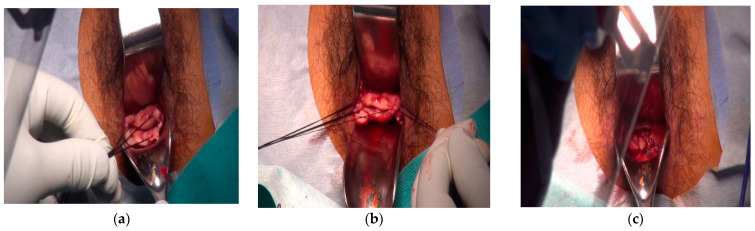
Vaginal cuff creation was performed as described in a previous report [19]. (**a**) Pull the needle from the vaginal wall and inject adrenaline saline solution for 400,000 times; (**b**) Make an incision with Metz Mechenbaum 3 o’clock 9; (**c**) Peeling the vaginal wall with a thickness of ≥2 cm.

**Figure 2 cancers-14-04389-f002:**
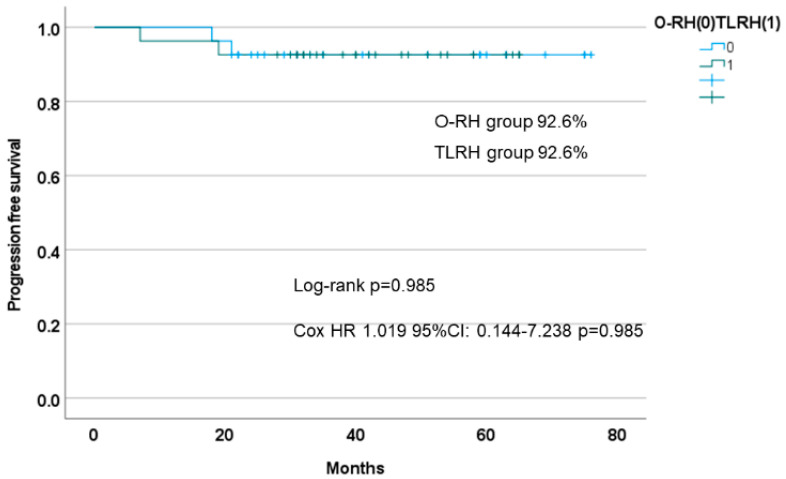
Kaplan–Meier curve for progression-free survival in propensity score-matched patients (TLRH vs. O-RH). O-RH: open-abdominal radical hysterectomy; TLRH: total laparoscopic hysterectomy.

**Figure 3 cancers-14-04389-f003:**
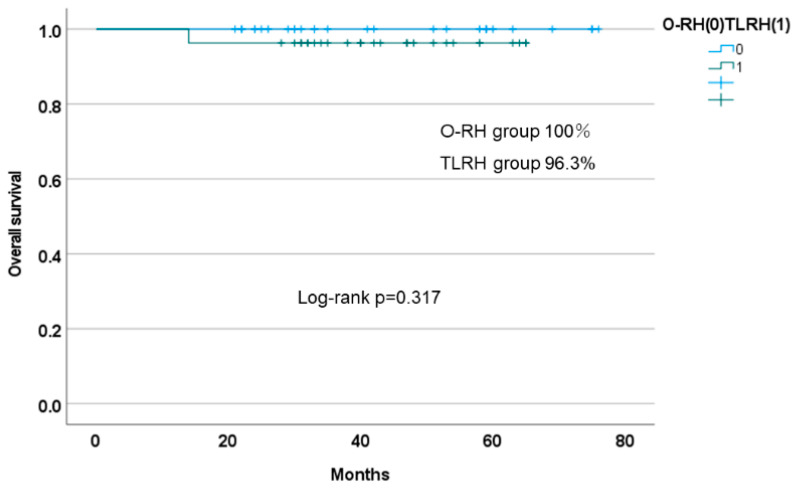
Kaplan–Meier curve for overall survival in propensity score-matched patients (TLRH vs. O-RH). O-RH: open-abdominal radical hysterectomy; TLRH: total laparoscopic hysterectomy.

**Figure 4 cancers-14-04389-f004:**
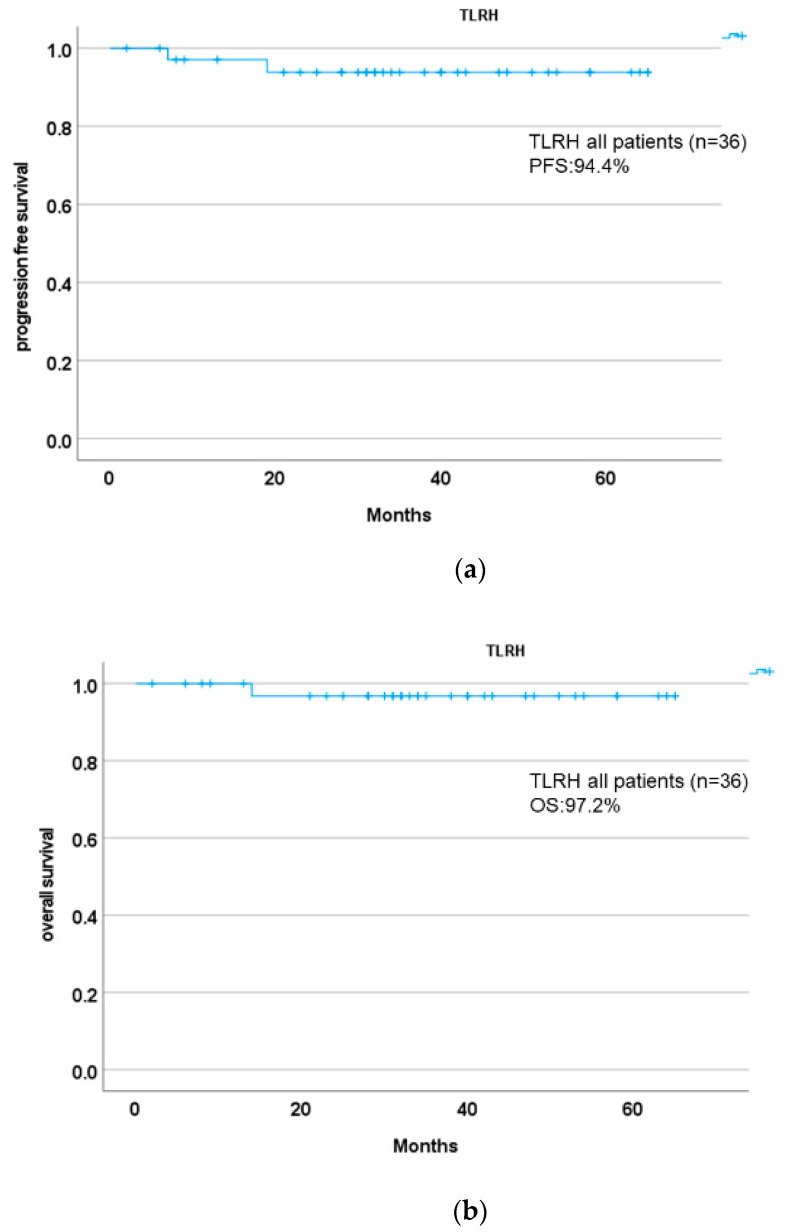
Kaplan–Meier curve for (**a**) progression-free survival and (**b**) overall survival in total laparoscopic radical hysterectomy in all patients.

**Figure 5 cancers-14-04389-f005:**
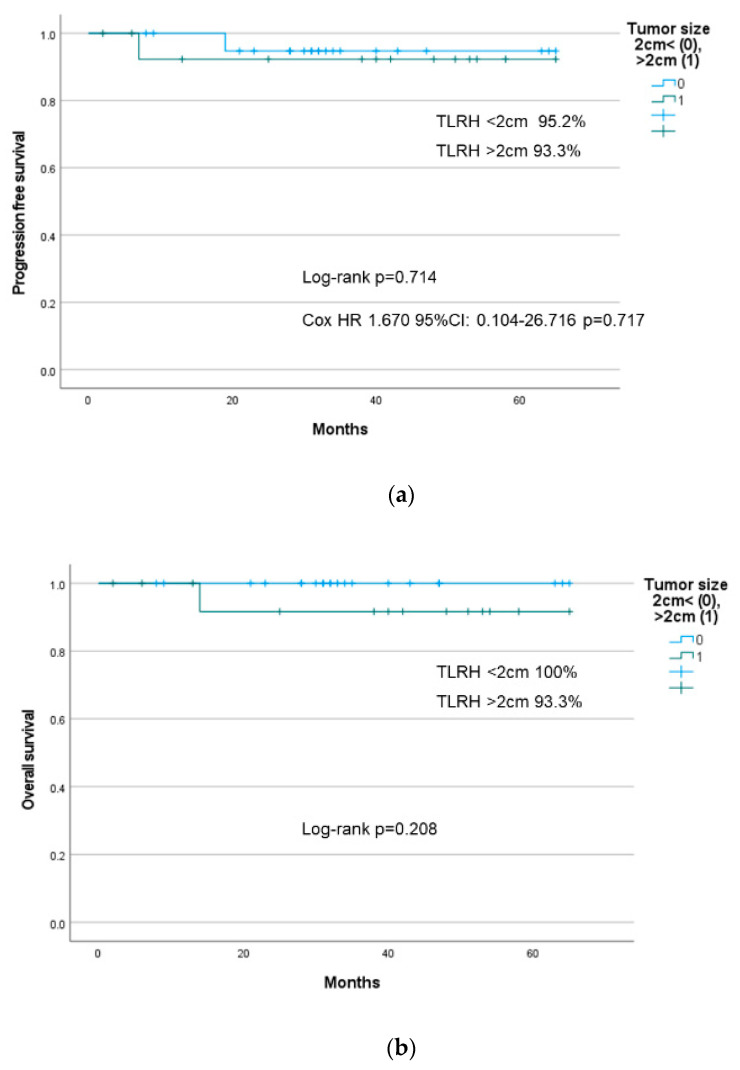
Kaplan–Meier curve for (**a**) progression-free survival and (**b**) overall survival in total laparoscopic radical hysterectomy (<2 cm vs. ≥2 cm).

**Table 1 cancers-14-04389-t001:** The background characteristics of stage 1B1-2 cervical cancer patients who underwent TLRH and O-RH.

	All (*n* = 94)	TLRH (*n* = 36)	O-RH (*n* = 58)	
Age, years				0.200
Median (range)	45.5 (28–77)	44.5 (28–77)	48 (29–76)	
BMI, kg/m^2^				1
Median, range	20.9 (15.1–35.4)	20.6 (17.9–27.8)	21.1 (17–35.4)	
Para				0.181
0	17	9	8	
>1	77	27	50	
Histology				1
SCC	70	27	43	
Adenocarcinoma/adenosquamous	24	9	15 *	
Conization				
Done		10 (27.7%)		
No conization		26 (62.3%)		
Tumor size (preoperative, MRI)				0.001
<2 cm	41	23	17	
2–4 cm	54	13	41	
Operative time (min)	382	382 (270–460)	382 (200–689)	0.768
Amount of bleeding	417	198 (5–1772)	517.5 (123–2360)	<0.001
Post-operative stage (FIGO 2018)				<0.01
1B1	34	20	14	
1B2	30	11	19	
2A1	3	1	2	
2B	2		2	
3C1p	25	4	21	
LVSI				0.03
+		16	39	
−		20	19	
Number of harvested lymph nodes	26 (12–63)	29 (18–57)	25 (12–63)	0.471
Lymph node metastasis				<0.01
+	25	4	21	
−	67	32	37	
Adjuvant therapy				0.08
Chemo	18	9	9	
Rad	5	2	3	
CCRT	32	6	26	
None	39	19	20	
Complication				
Blood transfusion		1	3	0.305
Ileus		0	0	-
Lymph edema (Grade < 1)		4	1	0.381
Lymphangitis (Grade < 2)		2	2	1.000
DVT (Grade < 2)		2	2	1.000
Cuff dehiscene (Grade < 1)		1	0	-
Port site hernia (Grade < 1)		1	0	-
Bladder injury (Grade < 1)		0	2	0.179
Obturator nerve injury (Grade < 1)		1	0	0.389

Basal cell carcinoma: one case. BMI: body mass index; SCC: squamous cell carcinoma; LVSI: lymphovascular stromal invasion; CCRT: concurrent chemoradiotherapy; DVT: deep vein thrombosis; O-RH: open-abdominal radical hysterectomy; TLRH: total laparoscopic hysterectomy.

**Table 2 cancers-14-04389-t002:** Propensity score matching of 54 cases, balanced for tumor size, post-operative stage, and lymph node metastasis.

	All (*n* = 54)	TLRH (*n* = 27)	O-RH (*n* = 27)	*p*
Age, years				
Median (range)	46 (29–77)	46 (30–77)	46 (29–75)	0.670
BMI, kg/m^2^				
Median, range	21.2 (15.1–35.4)	21.2 (17.9–27.8)	21.7 (17–35.4)	0.074
Para				0.022
0	8	7	1	
>1	46	20	26	
Histology				1
SCC	41	21	20	
Adenocarcinoma/adenosquamous	13	6	7	
Tumor size (MRI)				1
<2 cm	24	12	12	
≥2 cm	30	15	15	
Operative time (min)	382	384 (270–460)	377 (253–663)	0.768
Amount of bleeding	417	196 (5–1772)	550 (123–2360)	<0.001
Post-operative stage (FIGO2018)				0.224
1B1	39	15	12	
1B2	15	8	7	
2A1		1	1	
2B			1	
3C1p	11	3	8	
LVSI				1
+	27	13	14	
−	27	14	13	
Number of harvested lymph node	26.5 (13–57)	30 (18–57)	23 (13–55)	0.109
Lymph node metastasis				0.175
+	11	3	8	
−	43	24	19	
Adjuvant therapy				1.00
Chemo	10	7	3	
Rad	3	1	2	
CCRT	12	5	7	
None	29	14	15	

BMI: body mass index; SCC: squamous cell carcinoma; LVSI: lymphovascular stromal invasion; CCRT: concurrent chemoradiotherapy; DVT: deep vein thrombosis; O-RH: open-abdominal radical hysterectomy; TLRH: total laparoscopic hysterectomy.

**Table 3 cancers-14-04389-t003:** Analysis of collapsed vaginal cuff in the TLRH group.

Case	Age	Histology	Tumor Size (MRI)	Tumor Size (Histology)	LVSI	Recurrence	Rupture Site	Prognosis
1	47	SCC	35	45	-	-	Anterior	NED
2	44	SCC	27	25	+	+	Anterior	DOD
3	50	SCC	20	18	+	-	Rt, lateral	NED
4	40	AC	0	10	-	-	Lt, lateral	NED
5	74	SCC	16	9	-	-	Rt, lateral	NED

SCC: squamous cell carcinoma; AC: adenocarcinoma; LVSI: lymphovascular stromal invasion; NED: no evidence of disease; DOD: dead of disease; TLRH: total laparoscopic hysterectomy; MRI: magnetic resonance imaging.

## Data Availability

The data that support the findings of this study are available from the corresponding author, E.K., upon reasonable request.

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
