# Peer review of "Does Vaginal Cuff Creation and Avoidance of a Uterine Manipulator Improve the Prognosis of Total Laparoscopic Radical Hysterectomy for Early Cervical Cancer? A Retrospective Multicenter Study"

_cancers, 2022, doi:10.3390/cancers14184389_

Round 1
Reviewer 1 Report
The present study deals with a topic widely addressed in the literature and including much greater numbers and reaching the same conclusions. So I wonder what this study can add to current knowledge. A sample size calculation was not performed. I don't think the study's goal has been achieved.
Author Response
Response to the Reviewers’ comments
We thank the Reviewers for their valuable comments, which were very helpful for revising and improving our manuscript. We have reviewed the comments carefully and made the required corrections, which we hope will meet with your approval. The revisions are shown in red front and yellow marker in the updated manuscript.
Reviewer 1
The present study deals with a topic widely addressed in the literature and including much greater numbers and reaching the same conclusions. So I wonder what this study can add to current knowledge. A sample size calculation was not performed. I don't think the study's goal has been achieved.
âž¡Response
We have added the following discussion of the state of MIS for cervical cancer after the LACC trial to the Introduction (Page 2, Lines 84–98):
“After the LACC trial, a systematic review and meta-analysis by Nitecki et al. [12] reported that the pooled hazard of recurrence or death was 71% higher among patients who underwent minimal invasive radical hysterectomy compared with those who underwent open surgery (HR: 1.71; 95% CI, 1.36-2.15; p<0.001), and that the hazard ratio for death was 56% higher (HR: 1.56; 95% CI, 1.16-2.11; p=0.004). In contrast, a meta-analysis by Ronsini et al. [13] reported that laparoscopic-assisted vaginal radical hysterectomy is a safe MIS option, and that it does not appear to affect disease-free and OS in early stage cervical cancer patients. The MEMORY [14] have revealed that minimally-invasive radical hysterectomy for cervical cancer did not appear to compromise oncologic outcomes compared to open surgery, achieving similar PFS and OS rates. This may be due to the fact that these MIS procedures were performed by an experienced gynecologic oncologist. Furthermore, Alfonzo et al. [15] reported the safety of MIS for tumors >2 cm, while Manzour et al. [16] found no differences in patterns of relapse across surgical approaches in patients with stage 1B1 cervical cancer undergoing radical hysterectomy as the primary treatment.”
We agree with your comment that the small sample size is a limitation of our study. However, it is very important to report that MIS is a safe option and leads to similar outcomes to those of open surgery. Thus, our report is important for emphasizing that TLRH is a safe option with vaginal cuff creation and avoidance of a uterine manipulator.
Reviewer 2 Report
In this retrospective analysis 94 cervical cancer patients stage 1B1 were identified and compered according to the surgical method (open or minimally invasive). Through propensity score matching the authors claim no differenses regarding PFS as well as OS between the two groups.
The paper is in overall well written, the references appropriated used and the method adequately explained, inclusion the surgical procedure. In the post era of the LACC trial the study does add scientific values for the readers.
However some comments;
-Introduction line 83. There are reports such as Alfonzo E et al Eur J Cancer 2019, reporting safety of MIS with tumors >2cm that should be mentioned here.
Materials and methods
Overall well written and detailed
-Line 96: Are the included cases consecutive? Is there a local register in every hospital and if so how is the accuracy of the register?
Results
Line 127: Complications should be divided according to Clavien Dindo (ref Clavien PA et al Ann Surg 2009) to better compare with other studies.
Discussion
Line 323. The obvious difference between the groups such as higher post operative stage and lymph node spreading should be emphasized and problematized, even after propensity score matching. Shouldn't the higher post operative stage in the ORH group and lymph node spreading generate worse oncological outcome? The authors should acknowledge the imbalances between the groups in the limitations.
Conclusions
line 332. Given the retrospective nature of this study, the small sample size and the differences between the groups do not allow such a strong conclusion. Must be reformulated.
Author Response
Response to the Reviewers’ comments
We thank the Reviewers for their valuable comments, which were very helpful for revising and improving our manuscript. We have reviewed the comments carefully and made the required corrections, which we hope will meet with your approval. The revisions are shown in red front and yellow marker in the updated manuscript.
Reviewer 2
In this retrospective analysis 94 cervical cancer patients stage 1B1 were identified and compered according to the surgical method (open or minimally invasive). Through propensity score matching the authors claim no differenses regarding PFS as well as OS between the two groups.
The paper is in overall well written, the references appropriated used and the method adequately explained, inclusion the surgical procedure. In the post era of the LACC trial the study does add scientific values for the readers.
âž¡Response
We have added the following discussion of the state of MIS for cervical cancer after the LACC trial to the Introduction (Page 2, Lines 84–98):
“After the LACC trial, a systematic review and meta-analysis by Nitecki et al. [12] reported that the pooled hazard of recurrence or death was 71% higher among patients who underwent minimal invasive radical hysterectomy compared with those who underwent open surgery (HR: 1.71; 95% CI, 1.36-2.15; p<0.001), and that the hazard ratio for death was 56% higher (HR: 1.56; 95% CI, 1.16-2.11; p=0.004). In contrast, a meta-analysis by Ronsini et al. [13] reported that laparoscopic-assisted vaginal radical hysterectomy is a safe MIS option, and that it does not appear to affect disease-free and OS in early stage cervical cancer patients. The MEMORY [14] have revealed that minimally-invasive radical hysterectomy for cervical cancer did not appear to compromise oncologic outcomes compared to open surgery, achieving similar PFS and OS rates. This may be due to the fact that these MIS procedures were performed by an experienced gynecologic oncologist. Furthermore, Alfonzo et al. [15] reported the safety of MIS for tumors >2 cm, while Manzour et al. [16] found no differences in patterns of relapse across surgical approaches in patients with stage 1B1 cervical cancer undergoing radical hysterectomy as the primary treatment.”
-Introduction line 83. There are reports such as Alfonzo E et al Eur J Cancer 2019, reporting safety of MIS with tumors >2cm that should be mentioned here.
âž¡Response
We have accordingly cited Alfonzo et al’s 2019 study published in the European Journal of Cancer.
Materials and methods
Overall well written and detailed
-Line 96: Are the included cases consecutive? Is there a local register in every hospital and if so how is the accuracy of the register?
âž¡ Response
For all patients in this study, MRI findings were confirmed by multiple experienced physicians at meetings of the MGOS study group, and the diagnosis of cervical cancer 1B1 was made before surgery. We believe that this is an appropriate approach for registering cases.
Results
Line 127: Complications should be divided according to Clavien Dindo (ref Clavien PA et al Ann Surg 2009) to better compare with other studies.
âž¡Response
Thank you for your suggestion. We have accordingly reanalyzed the results using the Clavien-Dindo classification. Additionally, we have added the Clavien-Dindo Grade to Table 1.
Discussion
Line 323. The obvious difference between the groups such as higher post operative stage and lymph node spreading should be emphasized and problematized, even after propensity score matching. Shouldn't the higher post operative stage in the ORH group and lymph node spreading generate worse oncological outcome? The authors should acknowledge the imbalances between the groups in the limitations.
âž¡ Response
Thank you for your suggestion. While there were no significant differences after PSM, the patients in the laparotomy group had many poor prognostic factors. We have addressed this limitation in the revised Discussion as follows (Page 12, Lines 414–415):
“Fifth, even if there were no significant differences after PSM, the patients in the laparotomy group clearly had several poor prognostic factors.”
Conclusions
line 332. Given the retrospective nature of this study, the small sample size and the differences between the groups do not allow such a strong conclusion. Must be reformulated.
âž¡ Response
Thank you for your suggestion. We have accordingly revised the Conclusion as follows (Page 19, Lines 419–423):
“There was no significant difference in the prognosis between the TLRH and O-RH groups if the cervical cancer was stage 1B1 (FIGO 2008). Tumor spillage was prevented during surgery by the creation of a vaginal cuff and avoidance of a uterine manipulator. TLRH might be considered an effective procedure.”
Reviewer 3 Report
This is a very timely article given the interest in laparoscopic vs. open radical hysterectomies for cervical cancer. I think this is generally well written. I would have a couple of needed revisions prior to acceptance.
1. The staging should be updated to the most recent FIGO staging and reanalyzed given this.
2. Additional propensity matched patients would be included given a relatively small sample size chosen
3. The procedure for "vaginal cuff" is recapitulated from a prior article. Even in that article it is not well explained without additional visual images/videos. I would suggest a short video be included in supplementary attachments rather than similar pictures and the entire component where the technique is re-described verbatim from the prior article be removed and referenced
Author Response
Response to the Reviewers’ comments
We thank the Reviewers for their valuable comments, which were very helpful for revising and improving our manuscript. We have reviewed the comments carefully and made the required corrections, which we hope will meet with your approval. The revisions are shown in red front and yellow marker in the updated manuscript.
Reviewer 3
This is a very timely article given the interest in laparoscopic vs. open radical hysterectomies for cervical cancer. I think this is generally well written. I would have a couple of needed revisions prior to acceptance.
- The staging should be updated to the most recent FIGO staging and reanalyzed given this.
âž¡Response
We have revised the manuscript to use the FIGO stage 2018.
Therefore, we have changed the title from stage 1B to early cervical cancer,
in the revised Abstract (Page 1 , Lines 36, 37, Page 2 , Line 50), in the revised Discussion (Page 12, line 409-410)
- Additional propensity matched patients would be included given a relatively small sample size chosen
âž¡Response
Thank you for your suggestion. While there were no significant differences after PSM, the patients in the laparotomy group had many poor prognostic factors. We have addressed this limitation in the revised Discussion as follows (Page 12, Lines 414–415):
“Fifth, even if there were no significant differences after PSM, the patients in the laparotomy group clearly had several poor prognostic factors.”
- The procedure for "vaginal cuff" is recapitulated from a prior article. Even in that article it is not well explained without additional visual images/videos. I would suggest a short video be included in supplementary attachments rather than similar pictures and the entire component where the technique is re-described verbatim from the prior article be removed and referenced
âž¡ Response
In accordance with your comment, we have deleted the Figure 1 legend. In addition, we have added a supplemental video showing the vaginal cuff creation technique in the Material and Methods as follows (Page 5, Line 210)
Round 2
Reviewer 3 Report
The authors have responded to the major concerns for the study. It is acceptable in present form with minor editing required.
Author Response
Reviewer 3
This is a very timely article given the interest in laparoscopic vs. open radical hysterectomies for cervical cancer. I think this is generally well written. I would have a couple of needed revisions prior to acceptance.
- The staging should be updated to the most recent FIGO staging and reanalyzed given this.
âž¡Response
We appreciate the reviewer’s constructive suggestions on our manuscript.
The updated FIGO staging gives added importance to MRI as a method of accurately measuring tumor size.
Tumor sizes of all patients were diagnosed by MRI imaging in this study.
We have revised the manuscript to use the FIGO stage 2018.
We have revised the following sentence “TLRH group (n=37) was stage1B1 (n=20), 1B2 (n=11), 2A1(n=1), and 3C1p(n=4).
O-RH group (n=58) was stage 1B1 (n=14), 1B2 (n=19), 2A1 (n=2), 2B (n=2), and 3C1p (n=21) “in Table 1 Page 5-6.
Furthermore, we have revised the Table 2, Page 8..
We have added the FIGO stage for the recurrent patients in Result Page11, Line 309-314.
On analyzing the pattern of recurrence in the TLRH group, we found one case (FIGO stage IIIC1p) with ascending metastasis of the para-aortic lymph node. However, in the other case (FIGO2018 stage IB1), the tumor was disseminated to the peritoneum.In contrast, two cases of lung metastasis were observed in the O-RH group (FIGO2018 stage IB1 and IIB).
Finally,
We have revised from “stage 1B1-2 (FIGO 2018)” to “early” in Abstract Page2, Line 48, in Discussion Page 12, Line 334, and 338
- Additional propensity matched patients would be included given a relatively small sample size chosen
âž¡Response
Thank you for your suggestion.
Tumor size of patients in O-RH group was significantly larger than those of TLRH group. (p=0.001) in Table1.
We need to analyze the patients of 27 vs 27 to eliminate significant differences in patient background (p=0.224) in Table2.